# Fluoroscopic-Guided vs. Multislice Computed Tomography (CT) Biopsy Mode-Guided Percutaneous Radiologic Gastrostomy (PRG)—Comparison of Interventional Parameters and Billing

**DOI:** 10.3390/diagnostics14151662

**Published:** 2024-08-01

**Authors:** Michael P. Brönnimann, Jagoda Kulagowska, Bernhard Gebauer, Timo A. Auer, Federico Collettini, Dirk Schnapauff, Christian T. J. Magyar, Alois Komarek, Miltiadis Krokidis, Johannes T. Heverhagen

**Affiliations:** 1Department of Diagnostic, Interventional and Paediatric Radiology, Inselspital, Bern University Hospital, University of Bern, 3010 Bern, Switzerland; jagoda.kulagowska@insel.ch (J.K.); alois.komarek@insel.ch (A.K.); miltiadis.krokidis@insel.ch (M.K.); johannes.heverhagen@insel.ch (J.T.H.); 2Department of Radiology, Charité-Universitätsmedizin Berlin, 13353 Berlin, Germany; bernhard.gebauer@charite.de (B.G.); timo-alexander.auer@charite.de (T.A.A.); federico.collettini@charite.de (F.C.); dirk.schnapauff@charite.de (D.S.); 3Clinician Scientist Program, Berlin Institute of Health at Charité-Universitätsmedizin Berlin, 10178 Berlin, Germany; 4Department of Visceral Surgery and Medicine, Inselspital, Bern University Hospital, University of Bern, 3010 Bern, Switzerland; ctj.magyar@gmail.com

**Keywords:** gastrostomy, fluoroscopy, radiation dosage, radiography, interventional, tomography, cost analysis, operative time

## Abstract

Background: This study investigated and compared the efficacy, safety, radiation exposure, and financial compensation of two modalities for percutaneous radiologic gastrostomy (PRG): multislice computed tomography biopsy mode (MS-CT BM)-guided and fluoroscopy-guided (FPRG). The aim was to provide insights into optimizing radiologically assisted gastrostomy procedures. Methods: We conducted a retrospective analysis of PRG procedures performed at a single center from January 2018 to January 2024. The procedures were divided into two groups based on the imaging modality used. We compared patient demographics, intervention parameters, complication rates, and procedural times. Financial compensation was evaluated based on the tariff structure for outpatient medical services in Switzerland (TARMED). Statistical differences were determined using Fisher’s exact test and the Mann–Whitney U test. Results: The study cohort included 133 patients: 55 with MS-CT BM-PRG and 78 with FPRG. The cohort comprised 35 women and 98 men, with a mean age of 64.59 years (±11.91). Significant differences were observed between the modalities in effective dose (MS-CT BM-PRG: 10.95 mSv ± 11.43 vs. FPRG: 0.169 mSv ± 0.21, *p* < 0.001) and procedural times (MS-CT BM-PRG: 41.15 min ± 16.14 vs. FPRG: 28.71 min ± 16.03, *p* < 0.001). Major complications were significantly more frequent with FPRG (10% vs. 0% in MS-CT BM-PRG, *p* = 0.039, φ = 0.214). A higher single-digit number of MS-CT BM-guided PRG was required initially to reduce procedure duration by 10 min. Financial comparison revealed that only 4% of MS-CT BM-guided PRGs achieved reimbursement equivalent to the most frequent comparable examination, according to TARMED. Conclusions: Based on our experience from a retrospective, single-center study, the execution of a PRG using MS-CT BM, as opposed to FPRG, is currently justified in challenging cases despite a lower incidence of major complications. However, further well-designed prospective multicenter studies are needed to determine the efficacy, safety, and cost-effectiveness of these two modalities.

## 1. Introduction

Patients with tumor-induced or neurogenic swallowing impairments often require enteral tube nutritional support. Enteral nutrition offers physiological and economic benefits compared to parenteral nutrition for long-term care [1]. Since the early 1980s, percutaneous techniques have been used as an alternative to nasogastric tubes [2]. Percutaneous radiologic gastrostomy (PRG) is a minimally invasive procedure that reduces patient burden, can be performed under local anesthesia, and generally results in lower complication rates and mortality compared to surgical methods [2,3]. It also has advantages over percutaneous endoscopic gastrostomy (PEG), including shorter procedure times, higher success rates, and fewer major complications [4,5].

PRG can be performed using fluoroscopy or increasingly with computed tomography (CT). Hu et al. [6] provided a direct comparison of these modalities, revealing that fluoroscopy-guided PRG (FPRG) had a significantly shorter procedural time (25.57 ± 5.99 min vs. 45.47 ± 8.98 min for CT-guided PRG) and a markedly lower radiation dose (by a factor of 95) but no significant differences in complication rates. Nonetheless, the literature suggests higher rates of both major (5.9–10% vs. 4%) and minor complications (7.8–23% vs. 7.7%) for FPRG compared to CT-guided PRG [2,5,7,8,9,10,11,12].

Recent advancements in CT technology, including the introduction of CT fluoroscopy (CTF) and multislice CT biopsy mode (MS-CT BM), have significantly reduced radiation exposure during interventions. Prosch et al. [13] demonstrated that MS-CT BM results in a tenfold lower radiation dose compared to CTF (*p* < 0.001). Despite these advancements, there is no consensus on the preferred imaging guidance method for gastrostomy procedures, and financial considerations are also a critical factor.

In light of these developments, this study aims to investigate and compare the efficacy, safety, radiation exposure, and financial compensation of MS-CT BM versus FPRG for percutaneous radiologic gastrostomy. The goal is to provide insights into optimizing radiologically assisted gastrostomy procedures and informing clinical and financial decision making.

## 2. Materials and Methods

### 2.1. Study Population

This retrospective study analyzed 172 successful percutaneous radiological gastrostomies (PRGs) performed at our university hospital from January 2018 to January 2024. Cases that significantly impacted procedure time—such as those deferred to the radiology department without a nasogastric tube, combined multislice CT biopsy mode (MS-CT BM) and fluoroscopy-guided PRG (FPRG), prior gastric bypass, non-compliance, need for general anesthesia, or incomplete data—were excluded to ensure a more accurate comparison between the groups (see Figure 1).

### 2.2. Baseline Evaluation and Preparation

Before the procedure, all adult patients underwent a clinical examination, including a detailed medical history and standard blood tests. Requirements for the intervention included an International Normalized Ratio (INR) value below 1.5 or a Quick value above 60%, a hemoglobin (Hb) level above 80 g/L, and a platelet count greater than 50 × 10^9^/L. All patients were required to fast for 6 h and have a nasogastric tube placed by the ward doctor prior to the procedure. Four board-certified, independent interventional radiologists—two with seven years of experience each and two with over 15 years of experience—performed the interventions. Each interventionalist independently selected the modality based on personal preference, case assessment, and device availability. A radiographer and an interventional radiology nurse were present during the entire procedure. The patient was positioned supine on the table. Room air (6 × 50 mL) was administered through the nasogastric tube via a luer lock syringe and a three-way stopcock before the intervention began. If there were no contraindications, intravenous butyl bromide (20 mg; Buscopan; Opella Healthcare Switzerland AG, Risch, Switzerland) was administered to reduce gastric motility. The Avanos Introducer Kit for Gastrostomy Feeding Tube, including four SAF-T-PEXY T-Fasteners and a Flocare^®^ Gastro Tube CH 14 (Danone Germany GmbH, Haar, Germany, was used for all procedures (see Figure 2). The patient’s skin was sterilized with 10% povidone-iodine. For local anesthesia, lidocaine (1%, maximum 30 mL; Lidocaine Streuli^®^ Pharma AG, Uznach, Switzerland) was applied subcutaneously and deeper into the stomach wall at each necessary step.

### 2.3. CT-Guided PRG Technique

This technology was implemented at our institute for the first time during the review period. A Toshiba Asteion 4SL CT scanner (Canon Medical Systems Corporation, Ōtawara, Japan) was used. The CT scout view was examined to ensure the stomach was sufficiently full. If deemed adequate by the interventionalist, a native upper abdominal CT scan was performed over the target area and reconstructed at 1 mm increments. The multislice CT biopsy mode-guided technique was used for subsequent standardized intervention steps with the following scan parameters: 120 kV, 25 mA s, rotation time 0.5 s, range 12 mm with 3 slices of 4 mm (see Figure 3).

The interventionalist exited the room for each image acquisition. Upon activation by the radiographer, three contiguous slices were captured to verify the needle position and make any necessary adjustments. Adaptive iterative dose reduction using three-dimensional processing (AIDR3D STD) was employed to reduce radiation exposure. Each needle end position and guide wire position were checked at least by MS-CT BM, and the patient was moved out of the gantry for each action. If there were uncertainties or if the puncture needle was not visible (e.g., due to significant respiratory excursion), a control CT scan could be performed at any time. Three gastropexies (2 cm apart) were mandatory before wire placement with an 18 G needle, dilatation, and tube insertion using an over-the-wire push-type method. Air was administered again (3 × 50 mL) directly before inserting the gastric tube to generate sufficient counterpressure. To check the correct position of the tube and detect early complications, a final upper abdominal CT was performed after balloon blocking with a maximum of 5 mL sterile water (Aqua ad iniectabilia Bichsel; Large pharmacy Dr. G. Bichsel AG, Interlaken, Switzerland) and administration of a luminal contrast agent (10 mL Iopamiro 300 diluted with sodium chloride in a 1:2 ratio; Bracco Suisse SA, Cadempino, Switzerland). Additionally, a metalline^®^ aluminum vaporized coating compress (Lohmann & Rauscher; St. Gallen; Switzerland) was used to cover the wound and secure the tube.

### 2.4. Fluoroscopy-Guided PRG (FPRG) Technique

Upper abdominal sonography was performed to visualize the position of organs relative to each other, gastric distention, and any larger vessels along the access route. The medial edge of the liver and spleen was marked on the patient’s skin. During the intervention, only the interventionalist and the interventional radiology nurse were present in the room, wearing lead aprons (0.25 mm lead equivalent) and a thyroid shield. The AXIOM Artis system from Siemens Healthcare, equipped with a flat panel detector and configured with the lowest feasible fluoroscopy frame rate and radiation settings, was used. The Siemens machine allowed for 3 pulses per second. Regardless of the placement technique, all procedures included at least one lateral view and multiple frontal views of the stomach. Every gastric puncture was performed under fluoroscopic guidance. Three successful gastropexies (2 cm apart) were mandatory before proceeding to the next steps. Before releasing the T-fasteners, it was ensured that the needle tip was within the stomach by either aspirating air bubbles or injecting a contrast medium. The gastrostomy tract was established at the midpoint of the T-fasteners using an 18-gauge needle. Before dilatation, it was demonstrated under fluoroscopy that the wire was in the antrum and that a backward loop had been formed to ensure stability. Immediately before inserting the gastric tube, air was administered again (3 × 50 mL) to create adequate counterpressure. At the end of the intervention, the balloon was blocked, and the correct final position of the gastric tube was confirmed by luminal injection of contrast medium under fluoroscopy.

### 2.5. Follow-Up

Vital signs and general condition were monitored on the ward for 6 h. No peri- or post-procedural antibiotics were administered. The patient remained fasting until the fluoroscopy-guided follow-up the following day. Only after this follow-up could the nasogastric tube be removed and the gastric tube used. Regular medical and surgical teams conducted follow-up care in the hospital, evaluating complications, feeding support adequacy, and tube outcomes at least every three months. Any issues or dysfunctions noted during follow-up were directed to the interventional radiology department. Patients experiencing tube-related problems were instructed to contact the radiology department directly via a 24-h telephone hotline provided in the nursing care booklet distributed to each patient following the procedure. The stitches of the suture anchors could be cut to skin level by the family doctor after 5 days, and a regular tube change was indicated after 3 months.

### 2.6. Data Evaluation

All procedures were reviewed by a resident with four years of experience and a board-certified interventional radiologist with seven years of experience. Neither of them performed or participated in any of these interventions. We recorded patient demographics, radiation exposure parameters (Dose Length Product (DLP)/Dose Area Product (DAP)), procedural time (in minutes), success rate, and complications. The respective dose product was converted into the effective dose (millisievert (mSv)) using accessible approximation formulas and calculators [14,15]. Procedural time was defined as the interval from the first to the last acquired image in the Picture Archiving and Communication System (PACS). Intervention-related information, such as complications, was collected from the electronic patient file from one day before to three months after insertion (regular tube replacement). Complications were categorized as minor (superficial wound infection, minor leakage, dislodgment, hematoma in the abdominal wall) or major (hemorrhage, peritonitis, wound infection, perforation/tear, aspiration, tube displacement, sepsis) according to Kandarpa et al. [16].

### 2.7. Billing Comparison

Tariffs were compared, and the most frequent, comparable, modality-specific examination (MFCE) at our institution was used to assess financial performance. Details are recorded in TARMED and retrieved from the current version 01.09.00, effective since 1 January 2018 (see Table 1). TARMED represents the tariff structure for billing outpatient medical services in Switzerland. The current slot time of the MFCE is fixed in our planning tool for the modalities. To more realistically depict effective procedural time, 10 minutes were generally added for preparation and post-processing, including room and patient positioning. According to internal pre-evaluation, the esophageal passage for fluoroscopy and the CT chest–abdomen–pelvis with contrast media for CT were considered as the MFCE.

### 2.8. Statistical Analysis

Patients were divided into two groups based on the modality used. Patient characteristics and interventional parameters were compared using the Mann–Whitney U test for continuous variables and Fisher’s exact test for categorical variables. The Kolmogorov–Smirnov test was used to test for normal distribution. Statistical analyses were performed using commercially available software (International Business Machines Corporation (IBM) Statistical Package for the Social Sciences (SPSS) Statistics for Windows, version 28; IBM, Armonk, NY, USA). A senior statistician from the Clinical Trials Unit (CTU) of the Faculty of Medicine of the University of Zurich was consulted for expert statistical advice. *p*-values < 0.05 were considered statistically significant.

## 3. Results

### 3.1. Study Population

After applying inclusion/exclusion criteria, our final cohort comprised 133 patients with 55 CT guidance and 78 FPRG (Figure 1). Age (63.49 years vs. 65.37 years, *p* = 0.365) and sex (female 43% vs. 57%, *p* = 0.842) did not differ significantly between groups (Table 2).

### 3.2. Complications

Major complications occurred significantly more during FPRG (*n* = 8, 10% compared to *n* = 0, 0% in MS-CT BM-guided PRG; value = 6.542, *p* = 0.039, φ = 0.214). Tube displacement (*n* = 4/8), followed by hemorrhage (3/8) and perforation/tear (1/8), were noted. All major complications occurred early and were detected either within the first few hours during close monitoring or, at the latest, during the fluoroscopy-guided follow-up examination the next day. Only the gastric tear required surgical intervention with repair of the anterior stomach. The other complications were treated conservatively. The dislocated tubes were reinserted, and the bleeding stopped on its own after the tube was tightened. Minor complications did not differ significantly between both interventional modalities (12% (9/78) FPRG vs. 15% (8/55) CTPRG). A high rate of tube dislodgment occurred necessitating unscheduled reinsertion (9/17, *n* = 5 in FPRG and *n* = 4 in MS-CT BM-guided PRG). Minor leakage (3/17; *n* = 2 in FPRG and *n* = 1 in MS-CT BM-guided PRG), superficial wound infection (2/17; each *n* = 1), and hematoma of the abdominal wall (2/17; *n* = 0 FPRG and *n* = 2 in MS-CT BM-guided PRG) represented the remaining (Table 2). Two out of three minor leakages were resolved with a simple change and better tightening. For one minor leakage, a larger probe was inserted to seal it. Superficial wound infections were managed with local disinfection and more frequent dressing changes.

### 3.3. Interventional Parameters and Follow-Up

The effective radiation dose differed significantly and was, on average, about 65 times higher for MS-CT BM-guided PRG than for FPRG (*p* < 0.001). The procedural times also differed significantly, with FPRG being performed 41% faster on the mean (*p* < 0.001). Derived from the data of the interventionalists who have performed MS-CT BM-guided PRG the most since its implementation in our institute (person A, *n* = 21; person B, *n* = 20), several interventions in the higher, single-digit range (8–9 for linear and 6–10 for exponential learning curve shape [17]) were necessary to initially reduce the average procedural time by 10 min (Figure 4).

### 3.4. Billing Comparison

Both intervention modalities undercut the specified procedural time according to the current TARMED flat-rate tariff. A comparison with the most frequent comparable examination showed that only 4% of MS-CT BM-guided PRGs achieved the same level of reimbursement. A comparable deficit of CHF 572.11 was achieved in 89% of MS-CT BM-guided PRG cases (<60 min procedure time). In contrast, FPRGs were always the superior intervention modality (Figure 5 and Figure 6).

## 4. Discussion

PRG is a critical intervention for patients requiring long-term enteral nutrition, particularly when endoscopic methods are not feasible. The choice of imaging guidance—either FPRG or MS-CT BM-PRG—can significantly impact procedural outcomes. Despite the established utility of these techniques, there are limited comparative data on their relative efficacy, safety, and economics. Our study retrospectively analyzed 133 patients who underwent either FPRG or MS-CT BM-guided PRG. We found significant differences between the modalities in terms of effective dose, procedural times, and complication rates. MS-CT BM-guided PRG was associated with a substantially higher effective dose (65 times on average, *p* < 0.001) and longer procedural time (1.43 times on average, *p* < 0.001) compared to FPRG. Major complications were more frequent in FPRG (*p* = 0.039), while minor complications did not significantly differ between the two. Although procedural time could already be reduced after a higher, single-digit number of MS-CT BM-guided PRGs, FPRG demonstrated a more favorable financial performance compared to MS-CT BM-guided PRG and MFCE.

Our complication rates are consistent with the limited CTPRG literature and numerically superior FPRG data. For the first time, our study shows a significant difference in the occurrence of major complications (0% in MS-CT BM-guided PRG vs. 10% in FPRG). Tamura et al. [4] reported a higher incidence of major complications, which may be attributed to the less precise needle guidance when the patient remains in the gantry, compared to our approach where the patient was moved out for each imaging step. Additionally, they did not specify whether an initial CT scan was performed to evaluate vital structures, unlike our study. This might explain the 4% hemorrhage rate observed in their study, as CTF images typically have lower resolution. The most frequent major complication in FPRG, tube displacement (5.13%), aligns with findings from Yang et al. [12] and Mildenberger et al. [2]. We agree with Hu et al. [6] that fluoroscopy’s main disadvantages include difficulty in accurately determining anatomical relationships and lack of depth/angle estimation. Similarly, Lang et al. [18] found that major complications often stem from technical challenges and limitations of imaging guidance. Our most common minor complication, tube dislodgment (7.27% in MS-CT BM-guided PRG and 6.4% in FPRG), aligns with de Baere et al. [19], Yang et al. [12], and Lorentzen [10]. Our experience supports de Baere’s thesis on the challenges of maintaining the tube and the risk of balloon perforation with expandable balloon gastrostomy tubes. We observed a decrease in this minor complication after placing a red warning sticker on the balloon catheter, indicating “Do not use-balloon access warning!”

Although newer CT scanners and interventional software can significantly reduce radiation exposure (a 94% reduction in patient-absorbed dose in CT fluoroscopic-guided interventions compared to conventional CT-guided interventions [4]), the effective patient doses remain notably high. Hu et al. [6] reported a 95-fold higher radiation dose for CTPRG compared to FPRG. Our results, with an increased effective radiation dose by a factor of 65, are consistent with this finding. The lower factor in our study may be due to Hu et al. [6] not reporting the settings of their conventional CT-guided modality and whether intermittent image acquisition was used. Standardizing the output of intervention parameters such as effective dose (mSv) and defining major and minor complications is essential for comparing future study results. Although there is no specific limit for effective dose exposure [20], our values should be interpreted in relative terms. For instance, the average effective dose of our MS-CT BM-guided PRG (10.95 ± 11.43 mSv) is comparable to that of a diagnostic CT abdomen [19]. We suggest that CT scans contribute significantly to the radiation dose, and efforts to reduce radiation exposure should be investigated further. Future studies should explore and compare intermittent image acquisition techniques (CT fluoroscopic-guided PRG vs. MS-CT BM-guided PRG), as CT-guided interventions are likely to increase.

Our results align with the limited literature available, particularly regarding procedural time. We found a mean procedural time of 28.71 ± 16.03 min for FPRG and 41.15 ± 16.14 min for MS-CT BM-guided PRG. Hu et al. [6] reported median procedural times of 25.57 ± 5.99 min for FPRG and 45.47 ± 8.98 min for CTPRG. Tamura et al. [5] observed an overall mean procedure time of 25.3 min (95% CI 23.7–26.8 min), noting a learning curve over time (Figure 4). More complex cases may have been scheduled for CT because interventionalists could choose the modality during the planning phase. Our study highlights that while “time is money” [21], the faster method is associated with a significantly higher rate of major complications. Barkmeier et al. [22] also concluded that FPRG should be the procedure of choice. We anticipate an increase in CT-guided PRG insertions, especially given the complexity of the patient population (e.g., post-bariatric surgery). Therefore, efforts must be made to significantly reduce procedural time to maintain profitability.

Our study has several limitations. First, adverse events were not comprehensively evaluated due to the study’s retrospective nature and its single-center design. Second, the effective dose could not be precisely determined, which is crucial for assessing potential radiation risks. Conversion factors are typically used to estimate it based on parameters like CTDI or DLP. However, these conversion factors are often based on standardized examination areas, such as the entire abdomen, which may not fully reflect each procedure’s specifics. Third, the billing comparison is based on the current version of the Swiss tariff system, which may not directly indicate the profitability of the modality.

## 5. Conclusions

Based on our experience from a retrospective, single-center study, the execution of a PRG using MS-CT BM, as opposed to FPRG, is currently justified in challenging cases despite a lower incidence of major complications. However, further well-designed prospective multicenter studies are needed to determine the efficacy, safety, and cost-effectiveness of these two modalities.

## Figures and Tables

**Figure 1 diagnostics-14-01662-f001:**
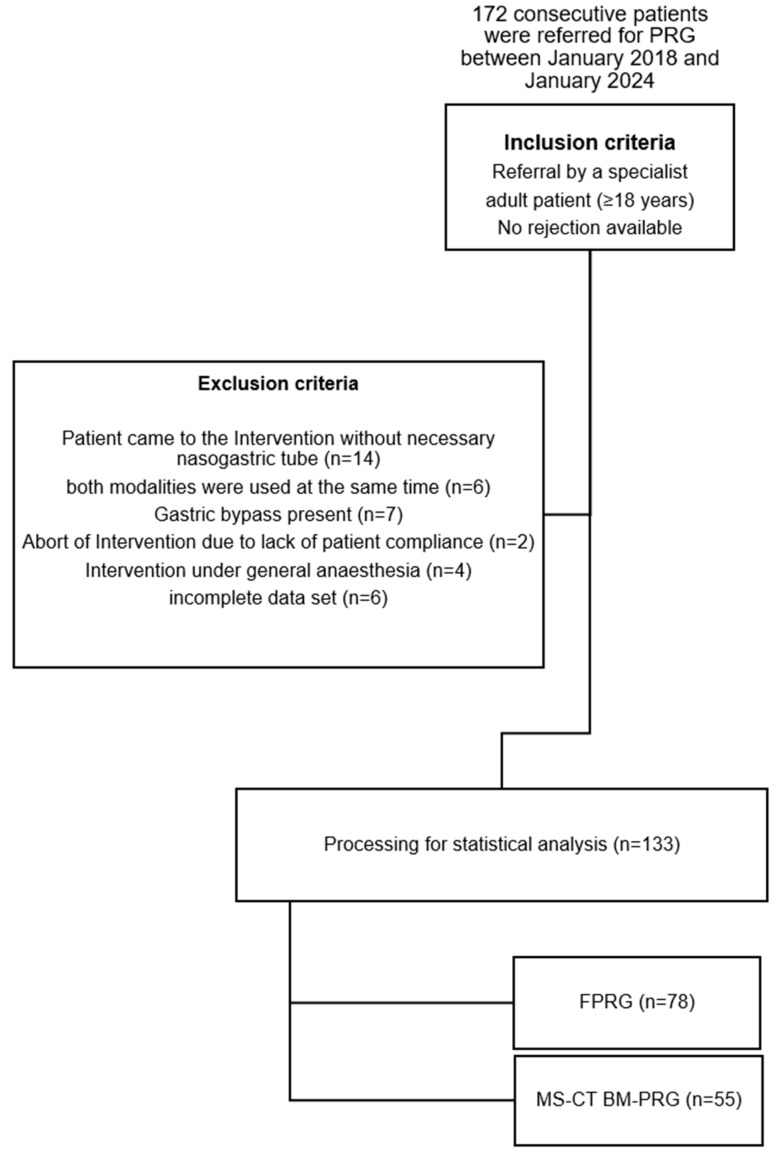
**Flowchart shows the study population.** FPRG = fluoroscopy-guided PRG; MS-CT BM-PRG = multislice computed tomography biopsy mode-guided PRG.

**Figure 2 diagnostics-14-01662-f002:**
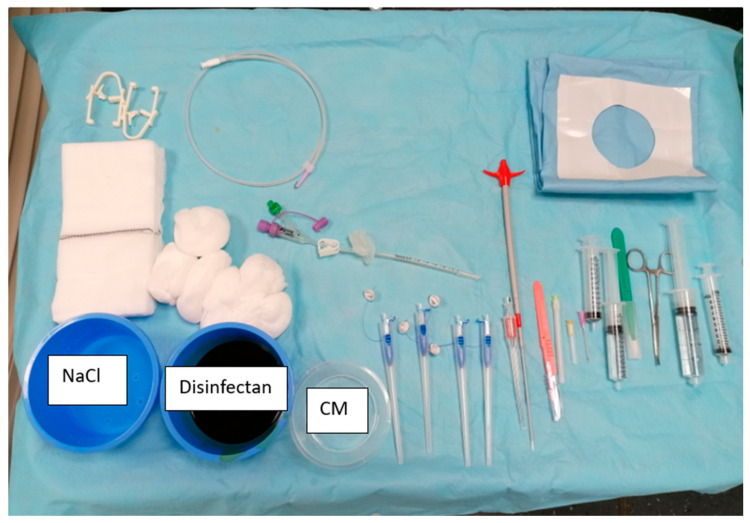
**Intervention material used.** Avanos Introducer Kit for Gastrostomy Feeding Tube with four SAF-T-PEXY T-Fasteners next to CM = contrast media, 10 mL lopamiro 300 diluted with sodium chloride with a ratio 1:2; Bracco Suisse SA, Cadempino, Switzerland; Flocare^®^ Gastro Tube CH 14 in the middle. A 0.035″ J-tip guidewire above the gastric tube. Lidocaine 1% in the bigger syringe. 10 mL sterile water in the other syringe, Aqua ad iniectabilia Bichsel; Large pharmacy Dr. G. Bichsel AG, Interlaken, Switzerland; NaCl = sodium chloride.

**Figure 3 diagnostics-14-01662-f003:**
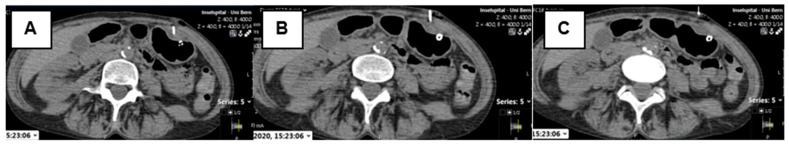
**MS-CT BM-guided PRG picture output.** Images of three sections are displayed after triggering by the radiographer or the foot pedal. The middle image (**B**) corresponds to the center where the gantry laser is visible. The first image (**A**) is cranial, and the last image (**C**) is caudal to this orientation point. This allows the needle position in space to be corrected. In this example, the puncture needle of the anchor runs from bottom to top with a slight lateralization. In contrast, CT fluoroscopy provides a real-time single image.

**Figure 4 diagnostics-14-01662-f004:**
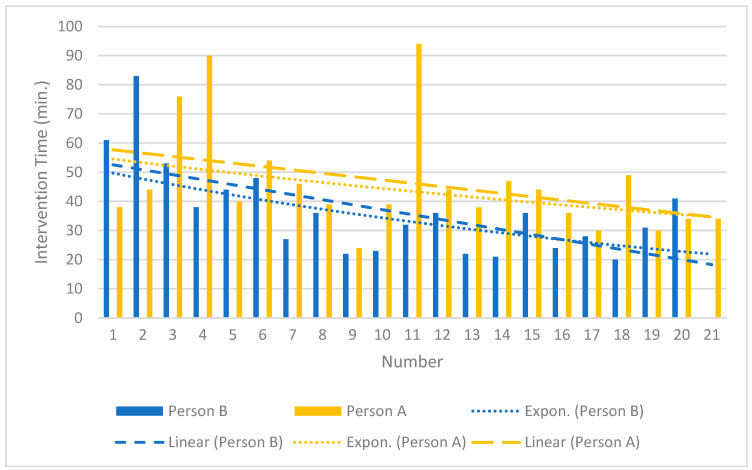
The learning curve of the MS-CT BM-guided PRG.

**Figure 5 diagnostics-14-01662-f005:**
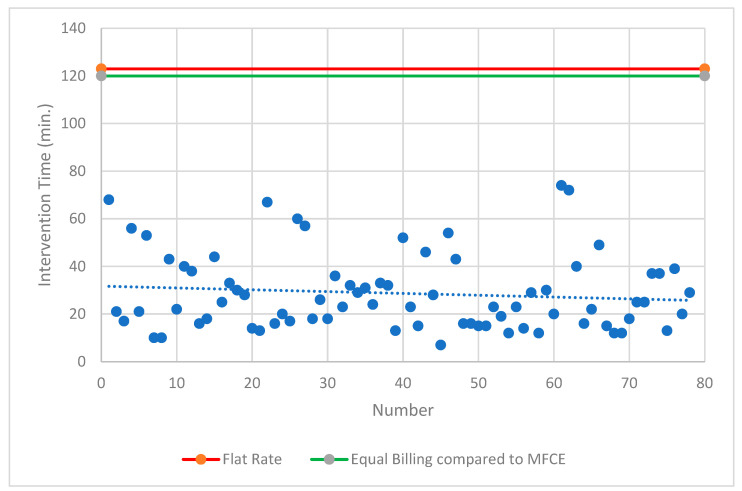
**Overview of the fluoroscopy-guided PRG.** Lines show the reimbursed flat rate for this intervention according to Switzerland’s TARMED Tariff structure and in comparison to the MFCE = most frequent comparable examination. PRG = percutaneous radiological gastrostomy; min. = minutes.

**Figure 6 diagnostics-14-01662-f006:**
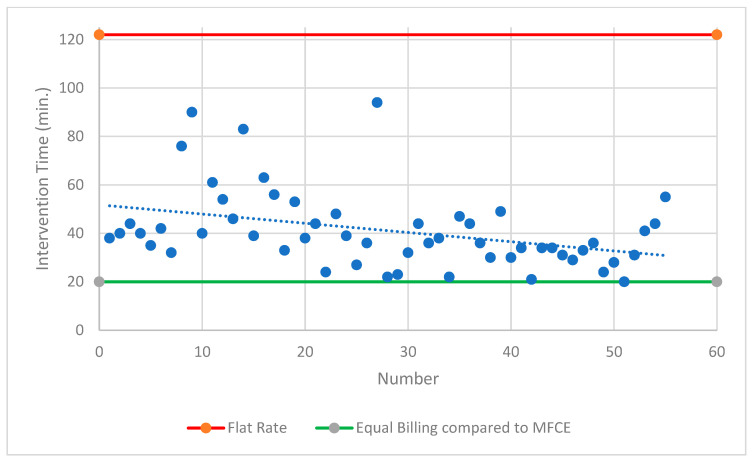
**Overview of the CT-guided PRG.** Lines show the reimbursed flat rate for this intervention according to Switzerland’s TARMED Tariff structure and compared to the MFCE = most frequent comparable examination; CT = computed tomography; PRG = percutaneous radiological gastrostomy; min. = minutes.

**Table 1 diagnostics-14-01662-t001:** Treatment/examination flat rates according to the TARMED Tariff structure for billing outpatient medical services in Switzerland.

TARMED Tariff Comparison				
	FLURO-MFCE	FPRG	MS-CT BM-PRG	CT-MFCE
	esophageal passage, OP			CT CAP, OP, CM
Total TP	354.25	1171.56	1024.04	945
Paid Min	undef	123	122	undef
IR Time (Mean IR Time + 10 min for PP and Doc		39	51	
Current Slot Time	30			20
Material Costs	included	498	498	included
Invoice Amount in CHF	294.46	1505.54	1378.67	975.39

FLURO = fluoroscopy; PRG = percutaneous radiological gastrostomy; FPRG = fluoroscopy-guided PRG; MS-CT BM-PRG = multislice computed tomography biopsy mode-guided PRG; MFCE = most frequent comparable examination; OP = outpatient; CM = contrast media; CAP = chest–abdomen–pelvis; TP = tax points; min = minutes; IR = intervention, PP = patient positioning and room preparation; Doc = documentation; CHF = Swiss francs.

**Table 2 diagnostics-14-01662-t002:** Patient demographics and lesion characteristics.

Survey of PRG Implantation							
Parameter	All (*n* = 133)		FLURO-Guided (*n* = 78)		CT-Guided (*n* = 55)		*p* Value
**Female**	35	26%	20	57%	15	43%	0.842
**Age (y)**	64.59	±11.91	65.37	±12.16	63.49	±11.59	0.365
**Effective Dose (mSv)**	4.629	±9.05	0.169	±0.21	10.95	±11.43	<0.001 *
**Intervention Time (m)**	33.85	±17.15	28.71	±16.03	41.15	±16.14	<0.001 *
**Complications**							0.039 *
major	8	6%	8	10%	0	0%	
minor	17	13%	9	12%	8	15%	
none	108	81%	61	78%	47	85%	

Unless stated otherwise, data are the number of implantations. X2 (2 x 2), X2 (R x 2), Fisher’s exact test, and the Mann–Whitney U test were used to calculate the statistical difference between groups of categorical, dichotomous, and continuous variables, respectively. Data are mean ± standard deviation. FLURO = fluoroscopy; PRG = percutaneous radiological gastrostomy; CT = computed tomography; y = year; m = minutes; * = significant.

## Data Availability

Dataset available on request from the authors.

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
