# Peer review of "Fluoroscopic-Guided vs. Multislice Computed Tomography (CT) Biopsy Mode-Guided Percutaneous Radiologic Gastrostomy (PRG)—Comparison of Interventional Parameters and Billing"

_diagnostics, 2024, doi:10.3390/diagnostics14151662_

Round 1

Reviewer 1 Report

Comments and Suggestions for Authors

Thank you for sending me this work for review. The work is beautifully prepared and presented. The results are also significant. I recommend that some revisions be made.

1-First of all, it is appropriate to review the study by a native English speaker in terms of English grammatical errors.

2- All abbreviations in the study should be explained. For example: hb, INR... like.

3-In the flow chart, 55 patients and 78 patients should be shown separately.

4-The most important part of this study is complications and should be explained in detail. What are the complications? How the complications that arose were treated. It should be presented as a subheading.

5-I recommend that the following work be examined in the discussion section.

1-Lang EK, Allaei A, Abbey-Mensah G, Zinn H, Walsh J, Derbes A, Bizzell C, Scalfani T, Nguyen Q, Raissi D. Percutaneous radiologic gastrostomy: results and analysis of factors contributing to complications. J La State Med Soc. 2013 Sep-Oct;165(5):254-9. PMID: 24350525.

6-In my opinion, in the conclusion part, it should be mentioned that there are few complications in the MS-CT BM method, especially the major complication.

Comments on the Quality of English Language

First of all, it is appropriate to review the study by a native English speaker in terms of English grammatical errors.

Author Response

General Statement:

Thank you very much for your very accurate work with exeptional effort. We hope that we have been able to fulfil your expectations.

Changes in detail:

  1. The entire manuscript was reviewed and corrected by the internal language revision team.
  2. missing abbreviations have been added: International Normalized Ratio (INR) and hemoglobin (Hb), page 3/12 lines 88/89; millisievert (mSv), p6/12 line 177
  3. The flowchart has been supplemented accordingly. P. 3/12, line 83.
  4. The complications were weighted accordingly in the results section and described in more detail. P 7/12, lines 226-242.
  5. The interesting work was included in the discussion section and acknowledged accordingly. P 10/12, lines 298-299.
  6. The conclusion has been clarified and expanded according to the proposed revisions. P.1/12, lines 37-41 and P.11/12, lines 344-349.

Reviewer 2 Report

Comments and Suggestions for Authors

I have read with great interest the manuscript by Brönnimann et al., entitled "Fluoroscopic-Guided vs. Multislice computed tomography (CT) Biopsy Mode-guided Percutaneous Radiologic Gastrostomy (PRG) – Comparison of Interventional Parameters and Billing".

This is an interesting retrospective study, that despite its limitations - already acknowledged by the authors - has scientific merit.

Few comments that need to be adrressed. 

ABSTRACT

in the CONCLUSIONS SECTION authors state " The execution of a PRG using MS-CT BM-PRG, as opposed to FPRG, is currently justified under challenging cases" GIVEN THE RETROSPECTIVE NATURE OF THE STUDY PLUS THAT IT WAS IN JUST ONE DEPARTMENT/HOSPITAL THE AUTHORS SHOULD REPHRASE BY ADDING "IN OUR HOSPITAL OR ACCORDING TO OUR EXPERTISE the execution of a PRG........... MORE WELL DESIGNED PROSPECTIVE MULTI-CENTRE STUDIES ARE REQUIRED TO DETERMINE THE EFFICACY, SAFETY AND ECONOMICS OF THESE TWO MODALITIES"

IN THE DISCUSSION SECTION, A SMALL INTRODUCTORY PARAGRAPH LEADING TO WHY THE AUTHORS UNDERTOOK THE STUDY IS MISSING.

IN THE CONCLUSIONS SECTIONS PLEASE SEE THE AFOROMENTIONED COMMENT IN THE ABSTRACT. 

Author Response

General Statement:

Thank you for your efficient analysis of our manuscript. We fully agree with your proposal and hope that we were able to realise your suggestions for improvement in accordance with your ideas and expectations.

Changes in detail:

1 and 3) The conclusion has been clarified and expanded according to the proposed revisions. P.1/12, lines 37-41 and P.11/12, lines 344-349.

2) This important addition has been included in a brief paragraph at the beginning of the discussion. P. 9, line 272-274 and P. 10/12, lines 275-276.

Round 2

Reviewer 1 Report

Comments and Suggestions for Authors

The authors fulfilled their responsibilities. The study can be published.